

# Technical note: A prototype transparent-middle-layer data management and analysis infrastructure for cosmogenic-nuclide exposure dating

Greg Balco[1]

[1]Berkeley Geochronology Center, 2455 Ridge Road, Berkeley CA 94550 USA

**Correspondence:** Greg Balco (balcs@bgc.org)

**Abstract.**

Geologic dating methods for the most part do not directly measure ages. Instead, interpreting a geochemical observation as a geologically useful parameter – an age or a rate – requires an interpretive middle layer of calculations and supporting data sets. Both of these are the subject of active research and evolve rapidly, so any synoptic analysis requires complete, repeated recalculation of ages from a growing data set of raw observations, using a constantly improving calculation method. Many important applications of geochronology involve regional or global analyses of large and growing data sets, so this characteristic is an obstacle to progress in these applications. This paper describes the ICE-D database project, a prototype computational infrastructure for dealing with this obstacle in one geochronological application – cosmogenic-nuclide exposure-dating – that aims to enable visualization or analysis of large, diverse data sets by making middle-layer calculations dynamic and transparent to the user. An important aspect of this infrastructure is that it is designed as a forward-looking research tool rather than a backward-looking archive: only observational data (which do not become obsolete) are stored, and derived data (which become obsolete as soon as the middle-layer calculations are improved) are not stored, but instead calculated dynamically at the time data are needed by an analysis application. This minimizes "lock-in" effects associated with archiving derived results subject to rapid obsolescence, and allows assimilation of both new observational data and improvements to middle-layer calculations without creating additional overhead at the level of the analysis application.

## 1 Interpretive middle layer calculations in geochronology

Geologic dating methods, saving a few exceptions like varve or tree ring counting, do not directly measure ages or timespans. Instead, the actual observation is typically a geochemical measurement, like a nuclide concentration or isotope ratio. Interpreting the measurement as a geologically useful parameter such as an age or rate then requires some sort of calculation and a variety of independently measured or assumed data such as radioactive decay constants, initial compositions or ratios, nuclide production rates, or nuclear cross-sections (Figure 1). This presents a problem for management and analysis of geochemical data because the interpretive middle layer between observable data and geologically useful information constantly changes with improvements in the calculation methods and new measurements of the other parameters needed for the calculation. Even though the geochemical measurements themselves in archived or previously published studies are valid indefinitely, the





derived ages become obsolete. This is an obstacle for analysis of geochronological data collected over a long period of time or, sometimes, from multiple laboratories or research groups who have different opinions about the middle-layer calculations, because any comparison requires repeatedly recalculating all the derived ages from source data using a common method. This paper describes a prototype computational infrastructure for dealing with this obstacle in one geochronological application

– cosmogenic-nuclide exposure-dating – that is intended to enable synoptic analysis of large, diverse data sets by making middle-layer calculations dynamic and transparent to the user.

## 2   Middle-layer calculations in cosmogenic-nuclide exposure dating

Cosmogenic-nuclide exposure dating is a geologic dating method that relies on the production of rare nuclides by cosmic-ray interactions with rocks and minerals at Earth's surface. As the cosmic-ray flux is nearly entirely stopped in the first few meters

below the surface, the nuclide concentration in a surface sample is related to the length of time that sample has been exposed at the surface. This enables many applications in dating geologic events and measuring rates of geologic processes that move rock from the subsurface to the surface, or from the surface into the subsurface (see review in Dunai, 2010). The most common of these is "exposure dating" of landforms and surficial deposits to determine, for example, the timing of glacier and ice sheet advances and retreats (e.g., Balco, 2011; Jomelli et al., 2014; Johnson et al., 2014; Schaefer et al., 2016) or fault slip rates and

earthquake recurrence intervals (e.g, Mohadjer et al., 2017; Cowie et al., 2017; Blisniuk et al., 2010).

The observable data for exposure-dating applications are (i) measurements of the concentrations in common minerals of trace nuclides that are diagnostic of cosmic-ray exposure, for example beryllium-10, aluminum-26, or helium-3, and (ii) ancillary data describing the location, geometry, and physical and chemical properties of the sample. Interpreting these measurements as the exposure age of a rock surface is simple in principle: one measures the concentration of one of these nuclides, estimates

the rate at which it is produced by cosmic-ray interactions, and divides the concentration (e.g., atoms/g) by the production rate (atoms/g/yr) to obtain the exposure age (yr). It is much more complex in practice, because the cosmic-ray flux, and therefore the production rate, varies with position in the atmosphere and the Earth's magnetic field, and the production rate also depends on the chemistry and physical properties of the mineral and the rock matrix. Production rate calculations are geographically specific, temporally implicit (because the Earth's magnetic field changes over time), and require not only a

model of the cosmic-ray flux throughout the Earth's atmosphere, but an array of other data including atmospheric density models, paleomagnetic field reconstructions, nuclear interaction cross-sections, and others. In addition, production rate models are empirically tuned using sets of "calibration data," which are nuclide concentration measurements from sites whose true exposure age is independently known.

The interpretive middle layer for exposure-dating, therefore, includes physical models for geographic and temporal variation

in the production rate, numerical solution methods, geophysical and climatological data sets, physical constants measured in laboratory experiments, and calibration data. All these elements are the subject of active research: new production rate scaling models and magnetic field reconstructions are developed every 1-3 years, and several new calibration data sets are published





each year. The result of this continuous development is that nearly all cosmogenic-nuclide exposure ages in published literature have been calculated with production rate models, physical parameters, or calibration data sets that are now obsolete.

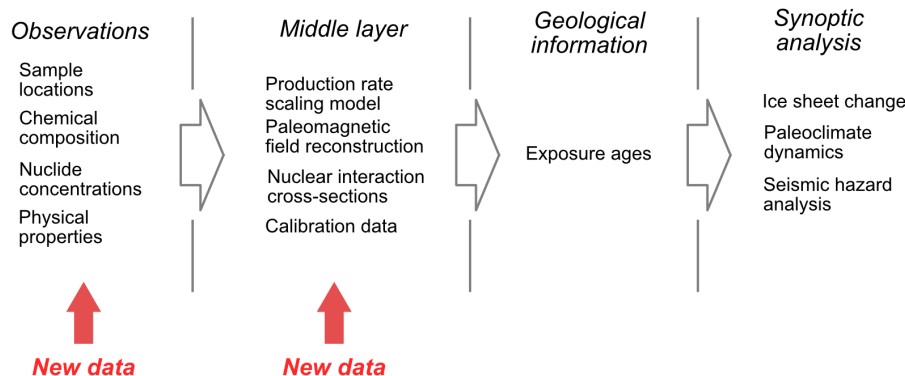

**Figure 1.** Conceptual workflow for applications of cosmogenic-nuclide exposure dating (or, in principle, nearly any other field of geochronology). Any large-scale analysis of ages or process rates needs to continually assimilate a growing observational data set and improving middle-layer calculations...or else it will be immediately obsolete.

It is rare (although possible) for middle-layer improvements to completely falsify or supersede the conclusions of previous research, but, regardless, any use of published data that are more than 1 or 2 years old, or any comparison of data generated at

different times or by different research groups, requires complete recalculation of exposure ages from the raw data. As there are tens of thousands of exposure-age measurements in the published literature, this is a major challenge to the use of these data for any sort of synoptic research. This is important because many of the most valuable uses of exposure dating involve large, geographically widespread data sets applied to, for example, analysis of regional and global glacier change (e.g., Young et al., 2011; Jomelli et al., 2011, 2014; Shakun et al., 2015; Heyman et al., 2016) or analysis of ice sheet change and sea level

impacts (e.g., Clark et al., 2009; Whitehouse et al., 2012; Nichols et al., 2019).

At present, middle-layer calculations for exposure dating most commonly utilize "online exposure age calculators" developed by, e.g., Balco et al. (2008), Ma et al. (2007), Marrero et al. (2016), or Martin et al. (2017), that consist of online forms accessible by a web browser into which one can paste sample information and cosmogenic-nuclide concentrations. The web server executes a script that carries out production rate and exposure-age calculations, and returns results formatted so as to

be easily pasted into a spreadsheet. The typical workflow for comparison or analysis of exposure-age data relies on manual, asynchronous use of one or more of these services, in which researchers: (i) maintain a spreadsheet of their own and previously published observational/analytical data; (ii) cut-and-paste from this spreadsheet into an online calculator; (iii) cut-and-paste calculator results back into the spreadsheet; and (iv) proceed with analysis of the results. Although the ability to use the online calculators in this way to produce an internally consistent set of results has been valuable in making synthesis of large data sets

drawn from multiple sources possible at all, this procedure creates redundancy and inconsistency among separate compilations





by many researchers, relies on proprietary data compilations that are, in general, not available for public access and validation, interposes many manual data manipulation steps between data acquisition and downstream analysis, creates a "lock-in" effect in which the effort required to recalculate hundreds or thousands of exposure ages using one scaling method is a disincentive to experimenting with others, and makes it difficult to assimilate new data into either the source data set or the middle layer

calculations.

## 3 A transparent-middle-layer infrastructure

These disadvantages of the current best-practice approach of manual, asynchronous use of the online exposure age calculators could be corrected, and synoptic visualization and analysis of exposure-age data better enabled, by a data management and computational infrastructure having the following elements.

1. A data layer: a single source of observational data that can be publicly viewed and evaluated, is up to date, is programmatically accessible to a wide variety of software using a standard application program interface (API), and is generally agreed upon to be a fairly complete and accurate record of past studies and publications, beneath:

   2. A transparent middle layer that calculates geologically useful results, in this case exposure ages, from observational data using an up-to-date calculation method or methods, and serves these results via a simple API to:

3. An analysis layer, which could be any Earth science application that needs the complete data set of exposure ages for analysis, visualization, or interpretation.

The key property of this structure is that only observational data (which do not become obsolete) are stored. Derived data (which become obsolete as soon as the middle-layer calculations are improved) are not stored, but instead calculated dynamically at the time the data are requested by an analysis application. This eliminates unnecessary effort and the associated lock-in

effect created by manual, asynchronous application of the middle-layer calculations to locally stored data by individual users, and allows continual assimilation of new data or methods into both the data layer and middle layer without creating additional overhead at the level of the analysis application. Potentially, this structure also removes the necessity for redundant data compilation by individual researchers by decoupling agreed-upon observational data (which are the same no matter the opinions or goals of the individual researcher and therefore can be incorporated into a single shared compilation) from calculations or

analyses based on those data (which require judgements and decisions on the part of researchers, and therefore would not typically be agreed upon by all users). The subsequent sections of this paper describe the ICE-D (Informal Cosmogenic-Nuclide Exposure-age Database) infrastructure, a prototype implementation of this concept.

## 4 The ICE-D implementation

The ICE-D transparent-middle-layer infrastructure prototype consists of (i) a networked database server storing observational

data needed to compute exposure ages, (ii) a networked Linux server that performs middle-layer calculations with MAT-



LAB/Octave code used in version 3 of the online exposure age calculator described by Balco et al. (2008) and subsequently updated, and (iii) a web server that responds to user requests by acquiring data from the database server, passing the data to the middle-layer server for calculation of exposure ages, and returning observations, derived exposure ages, and some related interpretive information to the user (Figure 2). The effect is that a user interacting with the web server can browse and work

5    with large data sets of exposure ages, originally collected and published by many researchers over several decades, without the necessity of managing the data set or recalculating all the exposure ages using a common method. Data management and middle-layer calculations are transparent to the user, allowing focus on data visualization, discovery, and analysis.

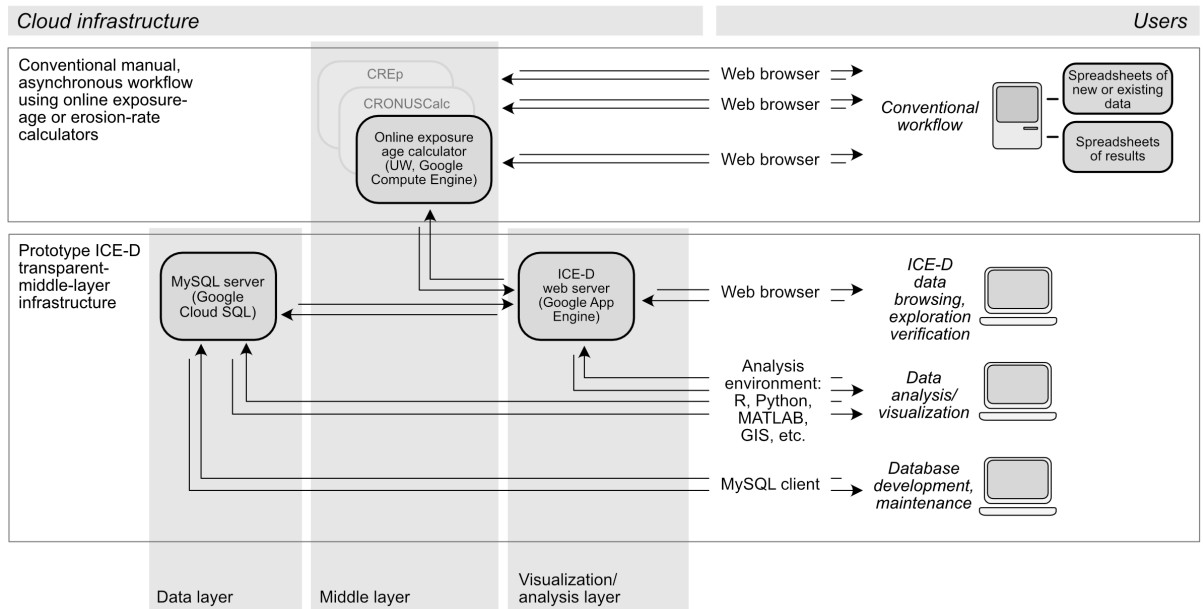

**Figure 2.** Generalized topology of the prototype ICE-D infrastructure compared to conventional manual, asynchronous use of online exposure age calculators. Cloud computing services interact with each other to supply raw data, calculated exposure ages, and other derived products to users at various stages of analysis.

The ICE-D prototype relies on cloud computing services available at low or zero cost from Google, Amazon Web Services, or other vendors; the current implementation uses Google Cloud Services (https://cloud.google.com). The data layer is a MySQL

10    database server provided by the Google Cloud SQL service. The middle-layer is a virtual machine, provided by the Google Compute Engine service, running CentOS 7 and the Octave code implementation of the online exposure age calculator, with a new API that facilitates programmatic use of the server. The web server that provides an example of a visualization/analysis layer is Python code running on the Google App Engine framework.





### 4.1 Aspects of the data layer

The purpose of the data layer is to store and serve observational data needed to calculate exposure ages, mainly including nuclide concentrations and the location, physical properties, and chemical properties of samples. It also includes some information useful for downstream analysis: for example, in a database containing exposure ages from glacial landforms, multiple samples from the same landform are grouped so as to signal that multiple ages can be averaged or otherwise combined to yield a better exposure age for the landform. The database has a standard relational database structure, with a series of tables containing information about landforms, samples collected from landforms, and geochemical measurements on samples. Additional data tables relate samples to publications, sources of research funding, and any digital resource with a URL (e.g., field and laboratory photos, detailed reports of laboratory analyses, etc.). It is similar to the database for cosmogenic-nuclide production rate calibration data already described by Martin et al. (2017).

In contrast to other services that aim to archive geochemical or geochronological data, the ICE-D database is not structured as a single entity designed to store any cosmogenic-nuclide exposure age data regardless of application, but instead consists of several separate focus area databases designed to contain restricted collections of exposure-age data needed for specific synoptic analyses. For example, ICE-D:ANTARCTICA (http://antarctica.ice-d.org) contains nearly all known exposure-age data collected from the Antarctic continent, the complete data set of which is important in reconstructing past changes in the extent and thickness of the Antarctic ice sheets. ICE-D:GREENLAND (http://greenland.ice-d.org) has a similar collection of used to reconstruct past changes in the Greenland Ice Sheet. ICE-D:ALPINE (http://alpine.ice-d.org) contains the majority of published exposure-age data from mountain glacier landforms worldwide, which in the aggregate are useful for paleoclimate reconstruction or diagnosis. The advantage of this focus-area approach is that developing relatively small (∼500 measurements for ICE-D:GREENLAND; ∼4000 for ICE-D:ANTARCTICA; ∼10,000 for ICE-D:ALPINE) data sets tailored to specific synoptic analysis applications enables a database project to become scientifically useful relatively quickly. The same number of measurements distributed among all possible global applications of exposure-dating research would likely result in many incomplete and not-particularly-useful data sets.

### 4.2 Aspects of the middle-layer calculations

The middle-layer calculations utilize version 3 of the online exposure age calculators originally described by Balco et al. (2008) and subsequently updated. Major improvements in version 3 in comparison to earlier versions described in the original paper include (i) an implementation of the production rate scaling method of Lifton et al. (2014) and Lifton (2016); (ii) a new API that returns exposure-age data as a compact XML representation rather than a web page, which facilitates programmatic use of the server, and (iii) many improvements in calculation speed relative to earlier versions and in comparison to other online exposure age calculators. The speed improvements are primarily derived from simple approximations for nuclide production by cosmic-ray muons (Balco, 2017) and extensive use of precalculated look-up tables instead of analytical or numerical formulae in the production rate scaling models.





### 4.3 Aspects of the prototype analysis and visualization layer

The prototype ICE-D web server is a simple example of the type of tool that could occupy the analysis and visualization layer. For the ICE-D:ANTARCTICA, ICE-D:GREENLAND, and ICE-D:ALPINE databases, the website provides a browse tree that allows one to view observational data and derived exposure ages for samples individually or grouped by, for example,

geographic region, landform, or publication. Views of samples or groups of samples include, in various combinations, a detailed report on observational data recorded in the database, exposure ages calculated using one or more production rate scaling methods, and some interpretive products such as analysis of the distribution of exposure ages on a particular landform (as is, for example, useful for glacial moraines in the ICE-D:ALPINE database) or age-elevation relationships for clusters of samples (as is useful for ice sheet thickness change reconstructions using the ICE-D:ANTARCTICA database). Thus, the current web

server implementation of the transparent-middle-layer concept replaces most aspects of the conventional practice of manual, asynchronous use of the online exposure age calculators, while also enabling continuous data assimilation and removing the need for each user to maintain a separate copy of the data set of keep exposure-age calculations up to date.

The prototype infrastructure also allows use of the transparent-middle-layer concept for many other analysis applications. For example, any analysis of exposure-age data, that would conventionally operate on a static, locally stored spreadsheet

or data file of previously calculated ages, can instead interact via standard APIs with the remote database and middle-layer servers to dynamically generate a data set of exposure ages at the time of analysis. Again, this allows the user to focus on the overall analysis and not on database maintenance and age recalculation tasks. In addition, if the analysis is structured as a program or script that acts on the current state of the database, rather than a one-time calculation in a static spreadsheet, the analysis can be continually updated to assimilate additions or improvements to the data layer and the middle-layer calculations.

For example, Balco (2020) showed some simple analyses of the age distribution of alpine glacier moraines worldwide: these analyses are performed by a MATLAB script that remotely queries the ICE-D:ALPINE database, so new data can be assimilated into the analyses simply by executing the script again. The prototype infrastructure also facilitates use of exposure-age data in geographic analysis applications. At present, the web server provides geolocated sample information in KML format to embedded map services that are displayed in web pages and used as a browsing interface, but it would also be possible to serve

both locations and derived exposure ages so as to allow viewing and analysis using desktop geographic information system software.

### 5 Social engineering aspects

An often noted obstacle to participation in community data management infrastructure (e.g, Fleischer and Jannaschk, 2011; Van Noorden, 2013; Fowler, 2016) is the conflict between the broad, generalized incentive for an overall research community

to develop centralized infrastructure, and the immediate incentives of researchers who might, for example, view individually authored publications as more critical to career-development objectives. The ICE-D prototype infrastructure has several features that could contribute to resolving this conflict. First, as discussed above, the separation of agreed-upon observational data from interpretive calculations or analysis makes the data compilation itself agnostic with respect to differences of approach or





opinion among researchers, thereby reducing potential disincentives to participation in database development. Second, from the perspective of an individual researcher, the transparent-middle-layer infrastructure can make it substantially faster and easier to carry out time-consuming or difficult tasks (e.g., statistical analysis, generating statistical or graphical comparisons of new and existing data, comparing data with model predictions) that are required to achieve individual goals (e.g., writing success-

ful proposals or publishing high-impact papers). In fact, more than 25% of sample descriptions in the ICE-D:ANTARCTICA database at this writing are unpublished data incorporated at the request of a number of researchers, and this may be evidence that the ability to use the analysis layer in tasks such as paper writing, proposal preparation, or sharing data with collaborators provides a positive incentive for user engagement with the project. User engagement with centralized data management systems should represent a trade – users provide a service to the community by making data available, and in exchange are provided

with services that help them to fulfill their own individual goals faster, better, and more easily. A transparent-middle-layer infrastructure can facilitate this exchange.

*Code and data availability.*  All data included in the ICE-D databases are publicly viewable via the respective websites (http://www.ice-d.org, http://antarctica.ice-d.org, http://greenland.ice-d.org, http://alpine.ice-d.org). Computer code for version 3 of the online exposure age calculators and the ICE-D web server is lodged in Google Cloud source repositories. Because no security evaluation has been conducted on

this code, read access is only available by request from the author. Note that this code is continually updated without versioning; the purpose of this paper is to describe the overall architecture of the system and not a specific version or snapshot of the code base.

*Competing interests.*  Balco is an editor of *Geochronology*.

*Acknowledgements.*  This work was supported in part by the Ann and Gordon Getty Foundation. Pierre-Henri Blard, Shaun Eaves, Brent Goehring, Jakob Heyman, Alan Hidy, Maggie Jackson, Ben Laabs, Jennifer Lamp, Alia Lesnek, Keir Nichols, Sourav Saha, Irene Schim-

melpfennig, Perry Spector, and Joe Tulenko helped develop the ICE-D:ANTARCTICA, ICE-D:GREENLAND, and ICE-D:ALPINE databases. Perry Spector as well as the Polar Geospatial Center at the University of Minnesota contributed to developing geographic browsing interfaces for both databases.





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
