# Peer review of "Technical note: A prototype transparent-middle-layer data management and analysis infrastructure for cosmogenic-nuclide exposure dating"

_Geochronology, 2020_

## Referee Comment (RC1) · Sebastian Kreutzer (Referee) · 31 Mar 2020

**Contribution summary**

The contribution presents and describes a prototype cyber-infrastructure to manage and analyse cosmogenic-nuclide dates termed *ICE-D*. Elements are databases, a computation environment, and an HTML-based graphical user interface accessible with any state-of-the-art web browser. To date, the project is organised into three sub-projects,

pooling datasets of related geographical origin. The system relies on a modular design concept, the author called this 'middle-layer'. Raw measurement data are separated from model assumptions and calculation tools used to determine exposure ages. The system does not contain age values, but measurement parameters to calculate ages on the fly.

**Recommendation**

I suggest a publication of this manuscript in GeoChron after a discussion of, what I believe there are, minor points.

**Justification**

What the author summarised in a brief technical note is quite a piece of software design and software development work. The manuscript is clearly written, logical structured, and it was my pleasure to read it. However, I have to admit that it is always hard to review such a manuscript. Software tools are moving targets, and what happens tomorrow with such a project, whether it gets accepted or rejected by the community, depends on various aspects; some of them are out of the author's control. The good news is that the paper did not bother the reader with a lengthy tool description, but focusses on some design aspects. The presented conceptional ideas are clearly of relevance beyond the cosmogenic-nuclide dating community. To me, the essential question the manuscript raises is *"How meaningful are data repositories storing final (exposure) age results?"*. The author tries to avoid a broad, potentially offensive discussion, but attempts an answer to that question by concluding that *"[published] [...] dataset [...] are now obsolete."*. Since I do agree with that reasoning, and in general with the presented concept, this review allows me to pick on some details, I highlighted

**GChronD**
while reading through the manuscript. Some of my comments the author may consider being more of major than minor nature, but I do not expect the author to rewrite the manuscript or go back to the workbench to refactor the software. What I want to start here is an open discussion, and some points may find their way into the revised manuscript.

**Detailed comments**

- 1. At first, I was struggling with the term 'middle-layer'. Finally, I thought 'yes', why not give the child a name. The author has chosen 'middle-layer' to express that particular calculation variable may evolve over time, and this may lead to different exposure ages. However, I would like to see a brief definition, somehow between line 16 and 24 on the first page, where the author says that "... therefore we decided to call 'middle-layer'...." or something similar. This would make it clear. Because what is presented here is not really new, it is something people did already in the past, and people do today when they align these ages (or at least they should do it).
- 2. At the end of section 1, I felt reminded to arguments already discussed in Wilkinson et al. (2016) who laid out the *The FAIR Guiding Principles for scientific data management and stewardship*. I think it would be good to add a few lines referring to their article. Because if data repositories and tools follow these guidelines, we would not have this problem.
- 3. Page 2, line 10: "...that a sample ... "
- 4. Page 2, line 12: "rocks" (?, plural instead of singular)
- 5. Page 2, line 17: I would prefer to see the chemical symbols instead of the names (the final decision is up to the author).
- 6. Page 3, first two lines: As written above, I am following your reasoning, however, as not being a member of the cosmogenic-nuclide community, I would like to see some numbers here, exemplifying the real effect. To which extent do exposure ages really change if specific parameters change? It does not need to be long, but it will make the manuscript stronger.
- 7. Page 4, your section 3: Maybe I've overlooked this aspect, but in this section, you talk about transparency and application interfaces etc. Where can I find the source code of this 'middle-layer' calculation (is it all part of the online calculator) in and how can I access the database without screencasting the webpage? If this is written already somewhere else, it should be repeated here. What I did expect from reading the manuscript, but before trying the webpage, was that the system comes with a dedicated API that allows other people to access the data the way they want it. This can even include the 'middle-layer', e.g., the API would enable using the middle layer calculator as a layer in between ... if wanted, if not, access should be possible directly to the database. After trying the webpage(s), I got the impression that this is not yet implemented, right? Please correct me if I am wrong and elaborate this a little bit in Sec. 3, because even it is a prototype, as I reader I want to get some idea about the project's future.
- 8. Page 5, Section 4: Here I have the same problem as above. Where can I see the source code, do I have an option to access those data directly? >> Below I found the option, but I kept the comment here to show that this was a question that crossed my mind while reading the paragraph.
- 9. Page 6, Sec. 4.1: I do understand why the author splits the system, I am not really in favour of this decision, but I don't have to. What I was missing on the webpage was a link back to the landing page where I can see all the sub-projects.
- 10. Page 5, the social engineering aspects: How users can contribute datasets? Do they get reviewed or accepted without a review? I certainly like to see tall these
data available, but why a user should contribute data to the project and what happens with them they decide, maybe later, that they don't want to have them anymore in the system?

- 11. Page 8, lines 12–16. I asked this two times above (sorry, I saw it just now) and here it comes, the code is available on request only, and the tool development is non-transparent. This aligns with the announcement that the system will be *"continually updated"*. The author should reconsider both decisions.
  - (a) These days agile software development goes hand in hand with rolling releases in conjunction with poor release versioning and an even less transparent recording of changes (if not developed via an open-source repository). Do we want to have this for scientific software tools? A proper and transparent (means visible to all users without having to place an inquire) bug and change tracking should be standard for scientific software solutions.
  - (b) Proper and transparent versioning of tools has the advantage that users can refer to it. This helps to track (potential) differences in the calculation. The author argues that the strength of the proposed solution is that ages are dynamically calculated, but maybe they don't have to be (re-calculated). Perhaps they had been all calculated with the same version or even with a different version, but the differences were related to cosmetic issues only (but we would not know). I understand that the calculator is somehow disconnected from the database and shows version numbers in the output, but also here. If the development was hosted in some kind of *git* repository (or similar), changes would be easily understandable.

**GChronD**
**Additional thoughts**

- 12. **Sustainability**: Recently, I read an interesting (non-scientific) blog entry about problems open-source software development projects face if the principal maintainer, here it appears to be the manuscript author, simply cannot continue the work. Who would take over? If everything would be available for download (or in a git repository), others could also help with the development and maintenance, and the system and all the work put in the past would sustain. After I saw that the webpage asks for donations to keep the service running (which is partly financed using private money from the author; my respect!), this issue the should think about.
- 13. **Usuability**: Is there a particular reason why I have to copy & paste the numbers to recalculate everything in the online calculator (either the exposure age calculator or *CREPp*, maybe this can be realised via sent button that fills the form?
- 14. Finally, I have two other questions that crossed my mind while testing the webpages (1) I could not find a proper Impressum, it would be good to add one since it will help to prevent legal problems and make clear who is running the page and for what kind of purpose. (2) Under what type of licence the datasets are published? No licence? Probably not, it seems that they can be used and re-used freely (there are in the database), but a licence should be clarified and added. Extrem case: Another group recycles the data and comes up with a completely new data interpretation based the conduced data mining. It should be clear from the start what kind of licence does apply if the data are available via *ICE-D*.

**GChronD**
**Conflict of interest**

I have no conflict of interest to declare. I am not a co-author or otherwise a beneficiary of the suggested references to be cited.

Sebastian Kreutzer - Bordeaux - March 31, 2020

**References**

Wilkinson, M.D., Dumontier, M., Aalbersberg, I.J., Appleton, G., Axton, M., Baak, A., Blomberg, N., Boiten, J.-W., da Silva Santos, L.B., Bourne, P.E., Bouwman, J., Brookes, A.J., Clark, T., Crosas, M., Dillo, I., Dumon, O., Edmunds, S., Evelo, C.T., Finkers, R., Gonzalez-Beltran, A., Gray, A.J.G., Groth, P., Goble, C., Grethe, J.S., Heringa, J., t Hoen, P.A.C., Hooft, R., Kuhn, T., Kok, R., Kok, J., Lusher, S.J., Martone, M.E., Mons, A., Packer, A.L., Persson, B., Rocca-Serra, P., Roos, M., van Schaik, R., Sansone, S.-A., Schultes, E., Sengstag, T., Slater, T., Strawn, G., Swertz, M.A., Thompson, M., van der Lei, J., van Mulligen, E., Velterop, J., Waagmeester, A., Wittenburg, P., Wolstencroft, K., Zhao, J., Mons, B., 2016. The FAIR Guiding Principles for scientific data management and stewardship. Sci. Data 3, 160018. doi:10.1038/sdata.2016.18

GChronD

---

## Referee Comment (RC2) · Sebastian Kreutzer (Referee) · 15 Apr 2020

Dear Dr Balco,

Thank you for this exciting and detailed response. It was more than I had expected, and it lays out compelling arguments and thoughts. To have a little bit of a discussion, I may respond to four points:

[Figure]

1. I still do not quite get the vulnerability argument that would require a cybersecurity audit first before the source code can be publicly available. This is not a NASA project. So, I guess this is more related to the way resources are hosted on external servers instead of servers at a research institute, which then imposes private liability you do not want to take (for understandable reasons). I am not part of the cosmogenic nuclide dating community, but this is obviously something the community should take up if they are interested in your project in the long run. I can only recommend that you try to host your project on severs part of a public research unit. Whether this is technically possible at all, of course, I cannot tell, and my comment should not give the impression that I hold this against you. Besides, maybe you can link to the 'API description' of the calculator? Ok, it is a blog entry, but it serves its purpose.

2. *[...] careful versioning of middle-layer code to facilitate reconstructing past calculations is fundamentally not consistent with the basic concept of storing only observational data and performing all calculations dynamically. If the middle-layer code is updated, then by definition the results of calculations using the new code are better than the results of calculations using the old code.*

   Yes and no. I have no doubt that every new version is released only with the best intentions to make the calculation better in terms of age accuracy and precision. However, it neglects the 'human factor'. Newer $\neq$ better and old $\neq$ wrong. Even with thorough manual and automated testing before the release, and I have no doubt that this is done, mistakes happen. The more it is essential to document and communicate this as soon as possible. Here transparent versioning helps a lot. Probably I spelt out the most apparent arguments you have heard a thousand times before. Perhaps our understanding collides simply because of a different perception of what users should do and what they actually do. You do believe,
and the manuscript leaves no doubt, that calculations should be done on the fly, always using the most updated version, which is immanent in the "middle-layer" concept. You have my full consent. However, my very personal experience with some software projects is that this is only remotely true to reality. People use data available and then add newly calculated data and start comparing them (without realising that they actually introduce a systematic error). Versioning would indicate whether this is safe to do so, or not. Probably this is my only point where I tend to say: It should be done, in particular given the minimum amount of additionally needed effort. I also think that this is not really off-topic when it comes to the manuscript, although such thing can be only implemented in the software prototype itself.

3. Regarding the rules for how data can be used. I realised that I was not clear enough. My intention was to provide some kind of "legal clarity" to users, data donators, and of course, the maintainer, you. Data can be harvested from published work but is it "safe" to store them again in an open repository? As long as a project does not raise too much attention, my best guess is that no one cares. But of course, the project should get attention, and it should be used. Such a project, as you have it presented, should take this into account by applying standard licences that allow legally safe reuse and recycling of data (e.g., creative commons licence) so far possible and applicable. Implementing such "rules" is not overly complicated and it does not restrain the user but avoids that a project with the bests intentions lashes back to people who started it and who use it. You use the word "rules". I do agree, creative minds don't like too many rules (and researchers are usually creative minds), and the feeling of being chained to something. However, what I was talking about was a minimum set of principles that helps to safeguard the valued degrees of freedom.

4. Yes, the FAIR-guidelines paint an ideal world and adhering them in whole might be impossible. I was asking for two to three additional lines of a reflection to

Interactive
comment

show that you are aware of it (it does not mean that you have to agree to all or endorse them), which I still think is justified and would strengthen the manuscript. Basically, a summary of your thoughts from the response would do. Why do I consider this as necessary? For example, the multi-billion euro European research framework programmes Horizon 2020 and Horizon Europe (hopefully from next year on), do not fail to emphasis over and over again the importance of those guidelines and make it even a prerequisite for projects' data management plans. The main reason for it might be that they have a lot of good ideas summarised and guidelines (in general) provide only guidance but do not set strict rules with the potential to lead to the opposite. Besides, I may have a distorted perspective because of my current position. Since I was never in favour of demanding the citation of a particular reference, I will finally leave this up to you to add a few lines (or not).

Finally, and just for the case that my comments leave doubts, my original recommendation did not and does not change, and I am pleased about this manuscript and the related discussion.

Sebastian Kreutzer – Bordeaux – April 15, 2020

---

## Author Comment (AC1) · 15 Apr 2020

This responds to the review comments by Sebastian Kreutzer. To summarize, this review is overall very supportive of the paper, but makes a number of minor suggestions to correct or clarify specific parts of the paper (review points 1, 3-5, 6, and 9), and also opens a discussion of several aspects of geochronology data management that are broadly related to the subject matter of the paper (points 2, 7,8, 10-14).

I very much thank Dr. Kreutzer for carefully reviewing the paper, and I am very happy

(perhaps too happy...the reviewer and everyone else may regret this by the time they get to the end of this response) to continue discussion on the topics he brings up.

I'll deal with the minor clarifications and corrections first. These include (i) a request to clarify the definition of "middle-layer" (review comment 1); (ii) a request for more discussion of how important changes to the middle-layer calculations are to the results of exposure-dating studies (comment 6); and (iii) several minor grammatical or style issues (comments 3,4,5). I agree with all these comments and will incorporate them into a revised manuscript. Finally, review comment 9 is a helpful suggestion for a minor change to the ICE-D website, but is unrelated to the text of the paper.

The remainder of the review does not propose changes to the paper, but instead includes a number of comments and questions about the ICE-D software infrastructure and highlights some broader implications of those questions for geochronology data management generally.

An important disclaimer for the following discussion of these broader comments is that (as noted in the 'competing interests' section of the manuscript) I am an editor of Geochronology. In this response, I am writing as a an author, not an editor, and my opinions on broader issues involving data management in geochronology should not be taken as any indication of journal editorial policy on these issues.

I think a good way of describing the context of the broader comments is that the review recognizes (and I agree) that there are a number of problems with how data management is currently handled in the field of geochronology, but this paper is only intended to address one of them. I agree completely with the points made in the review that (i) broader discussion of additional issues related to geochronology data management is valuable, but (ii) these issues are not pertinent enough to the main point of the paper to require specific revisions to the text.

I also appreciate and agree with the point at the beginning of the review that a peer-reviewed journal article might be the worst possible way to document a software system. The whole point of a journal article is to be an archive of information that is carefully checked and reviewed, and does not change after publication – but any useful software must evolve continuously as errors are corrected and functions are added or improved. If a software description in a journal article does not become obsolete rapidly, then the software is probably not very useful. This is the reason that the paper focuses on a conceptual description of the key elements of the software infrastructure and not on detailed documentation, and I am glad that the review recognizes this. Regardless, in the following paragraphs I'll try to answer most of the questions about details of the software implementation and the project philosophy, and give some broader perspective on why certain design decisions were made. I'll start with what I think are the more important or interesting issues and then descend into the minor points.

First, review comments 7, 8, 11, and 12 are questions about exactly how certain aspects of the software infrastructure works, and how one can use certain functions. The answer to most of these questions is going to be something like "Well, yes, but that's not really finished yet," and reflects the fact that the ICE-D software infrastructure is nothing like a complete, professionally developed, production system. It's a prototype that has some design features that I think are important and relevant to geochronology data management. As the review recognizes, the purpose of the paper is to describe these conceptual aspects and not to document the software in detail. In fact, throughout the paper I have been quite careful to describe it as a 'prototype' as often as possible, and not give the impression that any new aspects of the software are being presented as complete production tools.

One common part of the answer to all these questions has to do with the security of the various APIs and online interfaces. All contributors to the ICE-D project so far are Earth scientists with some self-taught knowledge of computational science, but no professional training or experience in software development. Clearly this is good enough to produce something that works OK most of the time, but it is unlikely to be good enough to produce software that is secure against malicious misuse. Hacking or
misuse is a serious risk for this project not only because it could destroy or corrupt data, but also because the system uses cloud computing services whose cost scales with usage, so a malicious takeover of ICE-D servers could potentially be very expensive. Given these circumstances, it seems unwise to permit public viewing of the source code. As noted in the 'access to data, etc.' section of the paper, there has been no security audit of this code by a competent professional, so I have no way to know whether the source code reveals vulnerabilities that could be used to hijack or misuse project resources. Clearly this is a problem that needs to be fixed, but it will have to wait until the project has enough resources to support a serious security audit. Until that time, really the only option is to restrict access to the source code to trusted viewers.

So, on to review comment 7. The source code for the various elements is in Google Cloud Source repositories, but is not publicly viewable, for the reasons discussed in the previous paragraph. The database is a standard MySQL server and can be accessed using any suitable client, but the server is firewalled to enable only specific IP addresses, and anonymous login is not permitted. I am happy to provide access to anyone who wants to use the database for research purposes, and at the time of this writing have done so for about 25 colleagues, collaborators, and students.

On the other hand, the online exposure age calculator web service API is publicly accessible, but is not very well documented, with the exception of some brief documentation here:

https://cosmognosis.wordpress.com/2014/09/26/a-web-service-implementation-of-the-online-exposure-age-calculator/

As discussed in the paper, the whole point of the project is that you should be able to plug into the infrastructure at any point – raw data, calculated ages, etc. – and all these capabilities do exist in prototype form and are used by the ICE-D web server. That is, all the arrows on Figure 2 have some real existence. However, public access at all levels is not currently feasible for security reasons, and documentation is very weak.

These are just facts imposed by the limited resources available to the project. Building a working prototype is one thing, but documenting all the APIs in detail and making them robust against misuse is a significant task that will require additional resources.

Review comments 8, 11, and 12: most of the same applies. I completely agree with the position that all the software should, in principle, be developed as public open source projects using normal version control and bug tracking systems. However, as discussed above, as a practical matter I don't think it's possible to meet ideal expectations right now.

Review comment 11 (b) does bring up one interesting side issue. Seen from the perspective of the overall project design, careful versioning of middle-layer code to facilitate reconstructing past calculations is fundamentally not consistent with the basic concept of storing only observational data and performing all calculations dynamically. If the middle-layer code is updated, then by definition the results of calculations using the new code are better than the results of calculations using the old code – so if one accepts this model in its entirety, there would never be any reason to want to reproduce obsolete calculations. Why would you want to reproduce calculations you know to be wrong? That is off topic from the perspective of this review and response, but is something to think about.

One final point that is important to keep in mind for the discussion of openness is that a poorly documented and only partially open system may not be ideal from the broadest possible perspective of open data and open science, but may still be extremely useful in facilitating scientific progress in a particular field. The ICE-D project has mainly developed up to this point as a collaboration infrastructure used by a relatively small number of cosmogenic-nuclide geochemists. It is not funded by any public agency. Although of course it would be great if it were also more broadly useful as a means of public access to data, and we have used a fairly open development model so far, there is no public access mandate, or any inherent expectation of usefulness outside a fairly restricted group of researchers. The immediate goal of the project is simply to facilitate
synoptic research using exposure-age data by creating a modern data management infrastructure. From this perspective, poor openness and poor documentation are not ideal, but also not necessarily critical disadvantages. If there are only, say, 20 people using the system, it is no problem to create 20 separate logins to the MySQL server, distribute the source code to 20 people, and personally explain how to use the APIs 20 times. Scalability is not needed. If the system enables new and useful Earth science research for those 20 people, this is a success. This is a very important broader point for this discussion: incremental progress is valuable in itself, and it is not necessary to develop a system that does everything for everyone in order to make major progress in computational infrastructure in a field. To return to the overall theme of this response, the point of this paper is to propose that the transparent-middle-layer infrastructure model contributes to solving one important problem in geochronology data management. The additional problems of public access, open-source software development, and good documentation are important, but they are different. Solving one problem is not as good as solving all problems, but it is a lot better than solving zero problems.

Points 10 and 14 in the review then address another much broader aspect of the ICE-D project. Essentially, these comments ask what the rules are for how data are incorporated into the databases, and what the rules are for how data can be used. Before discussing this subject, I want to reiterate the point that these issues are not particularly related to the main point of the paper that focuses on the transparent-middle-layer concept, so this discussion is largely unrelated to the paper itself. That being clear, however, for me this subject is one of the most interesting aspects of the overall project.

Basically, there are no rules.

The databases have been built and managed by a group of collaborators who have expertise in the field and interest in using the database (they are named in the acknowledgements section of the paper and on the websites). There is no formal review process, and members of this group correct errors in the data as they are discovered, using their best judgement to make sure that the observational data are correct and up

to date. There is no formal review or validation procedure for data ingest. The premise of the project is that its user group is a collaborative group of researchers working in the same field who all have a personal interest in ensuring that the data are complete and correct, and this approach leads to a "trust-but-verify" philosophy that requires a degree of trust on the part of users that the data set is a correct representation of past work, but allows users the opportunity to verify (or not) that trust by offering complete, granular, public viewability of the data.

There are also no rules for how data in the databases can be used. This is by design. First of all, the majority of data are drawn from published work and are indexed to publications, so are subject to whatever prior restrictions apply to the publications anyway, and additional restrictions would not be meaningful. In the ICE-D:ANTARCTICA database, about 80% of the data have been published elsewhere. The other 20% are unpublished data, and there are no rules as to how these data can be used. Unquestionably, this is unusual, because nearly all scientific data management projects are extremely concerned about who gets credit for what, and how people who generate data maintain "control" of data they have generated. Most of these projects have a lot of rules. In contrast, this has not been a concern in developing the ICE-D project. In the example of ICE-D:ANTARCTICA, a number of researchers, myself included, have incorporated large amounts of unpublished data in the database, even in the absence of any rules about who can do what with these data, because the perceived value of being able to use the transparent-middle-layer features of the database to collaborate with others, interpret their data, and compare it with other data sets, outweighs any perceived risks. Many other data management projects, mainly those focused on data archiving rather than on analysis tools, have assumed that researchers will be more likely to contribute to community data sets if there are more rules. That assumption might be wrong: rules themselves, by adding complexity and codifying an attitude of mutual distrust within a research community, may be a disincentive to participation. It may be much more effective to forget about the rules, which are mostly unenforceable anyway, and focus on highlighting the advantages, rather than the risks, of engagement with the project, making the incentives to participate strong enough to outweigh any disincentives.

Before this discussion travels too far from the main point, remember, again, that this subject has very little to do with the paper under review. The paper is about the application of a transparent-middle-layer software infrastructure to geochronology data management, not about the philosophy of who owns which data. For readers further interested in this subject, there is a much better articulated discussion of some of these aspects of the ICE-D project in a recent proposal that can be viewed here:

https://cosmognosis.wordpress.com/2020/02/25/computational-infrastructure-for-cosmogenic-nuclide-geochemistry/

Moving on to less significant points, review comment 13 is a question about the website user interface. The answer has to do with technical aspects of how the online exposure age calculators are typically used and, unfortunately, probably makes very little sense unless one is famiilar with them already. The text data string formatted for input to the online exposure age calculator is actually redundant on most of the pages served by the ICE-D web server, because the exposure age results that are commonly shown on the same pages are already the result of sending that data string to the online exposure age calculator. In other words, for most of the pathways of viewing data by site or sample, you don't need to cut and paste that text input string anywhere, because the web server already did that for you and is showing you the results. However, any user who is interested in computing exposure ages using an alternative production rate calibration would need the formatted input data. Thus, even though it's redundant, it appears on those pages to facilitate that workflow. Again, as noted above, the website has mostly been developed under the premise that the likely user group is already familiar with the exposure age calclator. Regarding the CREp formatting, at the moment CREp does not accept direct input of formatted text (or anything else) in an HTTP request, but instead requires uploading an Excel spreadsheet. Thus, going directly from the ICE-D web server to CREp is a coding pain in the neck that requires temporary file

writes and other overhead, and no one has invested the time needed to fix this problem yet.

And then, finally, review comment 2 calls attention to the "FAIR data principles." As background for readers who may not have heard of this, "FAIR" is an acronym intended to formalize and reinforce the common-sense approach that scientific (or any other) data should be "Findable, Accessible, Interoperable, and Reusable." In recent years this acronym has become popular in data management circles and associated with the Wilkinson et al. citation, although I am not sure if it originates there. From the perspective of this paper, certainly the transparent-middle-layer concept for managing and working with geochronology data has the potential to contribute to these goals: a centralized repository of observational data facilitates findability and accessibility; server implementations of both the database and the middle-layer calculations facilitate interoperability; and dynamic calculation of derived parameters from source data can enormously improve reuseability. On the other hand, there are several reasons that I did not mention this citation in the paper. The first one just relates to my earlier comment that there are lots of problems with data management in geochronology. This paper is only offering a solution to one of them, and makes no attempt to solve most of them; at no point does the paper claim that the ICE-D infrastructure is completely "FAIR." The second is perhaps more an issue of the philosophy of how citations should be used in scientific papers, but it seems to me that ascribing these basic concepts of information management, which probably date back centuries, to a 2016 paper is probably inappropriate. The third is getting fairly far off topic with regard to the present paper, and relates to the fact that in addition to establishing the acronym, the Wilkinson paper also articulates a set of "FAIR Principles" that include many prescriptive recommendations which go well beyond the basic ideas encapsulated in the acronym and seek to enforce a certain philosophy towards accessibility, metadata, and licensing. Although I agree with many of these recommendations in principle, my intention is to avoid endorsing them in the aggregate. In fact, I would argue that many of them are largely irrelevant in the context of a research tool primarily designed for use by

researchers in one field, and also that many of them may act to suppress, rather than advance, progress in data management by focusing on a theoretical ideal outcome rather than highlighting the value of incremental progress that may not be anywhere near the ideal, but is better than the status quo. To torture the common aphorism, I think judging improvements in data management against the "FAIR principles" often makes the perfect the enemy of the good, and geochronology data management is so far from perfect that we should be very happy about any incremental progress toward the merely good. This is a purely philosophical disagreement, and, again, I emphasize that my personal opinion of the "FAIR principles" is wildly off topic with respect to this review response – which also highlights that it is not very relevant to the paper and probably should not be brought up in the text. Following the disclaimer above, it is also important to make clear that my personal opinion of the "FAIR Principles" as expressed in this response should not be taken as any indication of journal editorial policy on this subject.

That's it. Again, I want to thank Dr. Kreutzer for his interest in this paper and his interest in discussing many of the issues raised in his review. However, although I agree that discussion of most of these issues is important in the broader context of geochronology data management, in this review process we should also keep in mind that the paper itself is intended to focus narrowly on only one issue.

―――――――――――――――

---

## Referee Comment (RC3) · Richard Selwyn Jones (Referee) · 26 May 2020

**Summary and recommendation**

The technical note describes ICE-D – a concept and computational infrastructure for managing cosmogenic nuclide exposure data. This prototype infrastructure is divided into three layers: data/observations, "middle-layer" age calculation, and analysis/visualisations. The new concept of this database is that the observational data is stored without the derived ages/rates, which are instead calculated based on up-to-

date methods when the user accesses the online database. To optimise data management and analysis, ICE-D has been divided into three region-specific databases: Antarctica, Greenland and Alpine. The author justifies the concept of this project, and describes the different components of the infrastructure.

First, I must congratulate the author for the time- and money-consuming development of ICE-D, which has already become a well-used platform for the cosmogenic nuclide community. More than just a database, the concept of observation-only storage with, what the author describes as, a "transparent-middle-layer" infrastructure has the potential to advance data management beyond the immediate research field. The topic is well within the scope of GChron, and the paper is well-structured, clearly and concisely written, with useful illustrations of the concept and infrastructure.

I recommend publication after consideration, and perhaps discussion, of points below.

**Detailed comments**

*The middle-layer calculations*

The use of the ICE-D database is dependent on the Balco et al. (2008) age calculator, which needs to be more explicitly addressed. There is nothing necessarily wrong with using this age calculator – in fact, the calculator is arguably as accurate as any other, and more efficient – but there are differences in calculation methods between the various online calculators, and even between the published version 1 (Balco et al., 2008) and the current version 3, which will lead to different resulting exposure ages. I appreciate that the author does not want to get into the details of the calculator, especially as some of this information will become outdated as the calculator updates to use new production rate information, scaling models, etc. However, I think that this paper – as well as the ICE-D website – should more clearly outline what methods/values are specific to this particular age calculator, and to what degree the ages may differ to those generated from other calculators.

Section 4.2 could be expanded to 1) outline in more detail the changes from version 1 to version 3, and 2) discuss the differences between this age calculator and the other available online calculators – for example, why might ages calculated with a different calculator but using the same production rate scaling method, get different exposure ages to that calculated within ICE-D?

There are also slight limitations to using this age calculator that should be acknowledged. For example, the user cannot get ages from ICE-D that are: calculated for a user-specified (local, or alternative global) production rate, as with CREp (Martin et al., 2017); or, calculated with the full range of observational data uncertainty, as with CRONUSCalc (Marrero et al., 2016) – correct me if I am wrong.

**Access to data**

In order for this database project to suit the needs of the community, the data will need to be accessible for use beyond the analysis and visualisation provided by ICE-D. The ICE-D online interface provides all of the available observational data, as well as dynamically-calculated exposure ages, in a fairly clear format and navigable structure. But this information could be made more accessible.

The information is not all stored on the same webpage, and it is assumed that the user should just copy the required information from the various tables. There are some limitations to this current set up. The tables are not always easy to copy-and-paste from the webpage to a local file (e.g. spreadsheet), particularly with the nuclide boxes. Not all sample information is available in a single table, meaning that the user has to move around the various locations to extract all of the data. All of the necessary sample information for age calculation is only available together in the format required for the Balco et al. (2008) version 3 calculator, and otherwise has to be found on each sample-specific webpage. What if a user wants to gather all sample information (observations necessary for independent age calculation and perhaps also ages provided by ICE-D) for a particular region, site or publication?

To overcome these limitations may require a substantial restructuring of the ICE-D online platform, and I leave it to the author to decide whether they want to make such changes. I have two suggestions:

1. Provide a table with all available sample information (including ICE-D dynamically-calculated ages) for data when grouped by region, site or publication. This could either be within the current page structure, as a link to a separate page with a table generated for that specific dataset, or as a downloadable file (e.g. tab-delimited .txt file) for that specific dataset.

2. Provide an option to extract user-specified sample information from ICE-D. This could be a link to a code package, or command line that could be entered on a local computing system. Either of these could be added to the data pages, again, when grouped by region, site or publication.

***Additional comment***

I also have slight concerns about the long-term maintenance of ICE-D and sustainability of the project – which is an issue for any such database project – but this has largely been addressed by the other reviewer.

---

## Author Comment (AC2) · 27 May 2020

This responds to the review comments by Richard Jones. Overall, this review is generally supportive of the paper and agrees that the transparent-middle-layer concept highlighted in the paper is a potentially useful contribution to geochronology. It then goes on to discuss a number of points that focus on aspects of the prototype implementation of this concept.

I very much appreciate Dr. Jones's close attention to these areas in writing the review.

[Figure]

These points are all valuable and worthy of discussion, and I discuss them below at some length, but they are also somewhat ancillary to the main point of the paper, which is to discuss the overall transparent-middle-layer concept. Thus, at the end of this response I will try to return the focus to what the implications of these areas of discussion are for actually revising and improving the paper.

Basically, the bulk of the review includes discussion of two main areas: (i) some aspects of the online exposure age calculator that forms the middle layer in the prototype implementation, and (ii) some suggestions for improving the web server application that forms the analysis layer of the prototype application.

The first of these mainly focuses on the fact that the online exposure age calculator used in the prototype (version 3 of the online exposure age calculators originally described by Balco and others (2008) and subsequently updated) is not very well documented. This is true and this comment certainly has merit. The original online exposure age calculator from 2009 set a high bar for documentation: not only was a paper published describing it, but there was probably a hundred pages of additional online documentation of mathematical formulae, details of numerical implementations, and comprehensive instructions for using the MATLAB code. The latest version, on the other hand, only has fairly general descriptions of basic concepts, and is nowhere near as well documented as the original. The fact is that as the current version is more or less a purely volunteer effort at this point, it has not been feasible to get to the same standard. There are several potential solutions to this problem, the simplest one most likely being to establish a documentation wiki that can be contributed to by multiple users of the calculator, so as to spread the responsibility more widely among interested and engaged users such as Dr. Jones. Of course, this has not been implemented either, but at least there is a fairly clear path for how to do so.

From the perspective of the paper being reviewed, however, the shortcomings of the documentation of the online exposure age calculator used in the middle-layer calculations are somewhat beside the point. The whole point of the distributed cloud infrastructure used for the prototype implementation described in the paper is that there is no reason that there has to be only one middle-layer server: in principle, any of the existing online exposure age calculators could fill this role. As a practical matter, however, right now there is only one that both (i) is fast enough for dynamic exposure age calculations and also (ii) has a programmatic interface that allows other software to easily submit input data and receive clearly formatted results. The other two options are the "CRONUSCalc" online calculator of Marrero and others (2016) and the CREp system of Martin et al. (2017). CRONUSCalc does not offer a programmatic API, is currently too slow for convenient dynamic calculations, and returns results by email instead of http or another more usual protocol. CREp also lacks a simple API, requiring instead a file upload for data submission. If these online calculators were to develop a simple http- or other standardized-protocol-based API in future, they could easily be used as middle-layer elements.

The second main point of the review focuses on a number of suggestions for improvements to the ICE-D web server that forms the visualization and analysis layer of the prototype implementation. First, Dr. Jones correctly points out that it would be extremely useful if the ICE-D web server duplicated the feature of the online calculators that non-default production rate calibrations could be propagated into the results. I agree completely, and this would be reasonably straightforward to develop, but I haven't done it yet. The reviewer then goes on to highlight a number of other potential functions of the ICE-D web server that would improve his workflow for data analysis, including presentation of data in various aggregated tablular forms as well as improvements to the HTML coding to facilitate copy-and-paste operations between browsers and spreadsheets.

These are all helpful and valuable suggestions, but again focusing on the present paper, two things are important here. First, the purpose of the paper is to highlight the overall usefulness of a transparent-middle-layer architecture by describing a prototype implementation that has a number of features designed to show proof of concept and

also to be a guide for what could be developed in future. Clearly this is working, because the simple capabilities that exist in the prototype have inspired this reviewer to come up with ideas for lots of other more advanced or more specific capabilities. Second, in the same way that many different online exposure age calculators could occupy the middle layer, the whole point of the distributed architecture is that there can also be many different software applications occupying the visualization and analysis layer. Dr. Jones could develop, or cause to be developed, his own analysis layer application with features designed for his specific workflow. Of course, in the current prototype situation, that is a bit difficult because the various APIs are mostly not fully documented, and it is still necessary to interact with me personally to get access permissions to the various server elements, but I would be very happy to help achieve it.

Finally, to refocus the discussion on improvements or modifications to the present paper that are indicated by this review, my take on this is that the main needed improvement is to include additional discussion and emphasis of the principle that although certain example middle- and analysis-layer elements are used in the prototype, the whole point of the distributed architecture is that it makes the transparent-middle-layer concept agnostic with respect to what elements are used. The specific middle- and analysis-layer elements used in the prototype are not themselves an inherent part of the transparent-middle-layer architecture. Multiple calculation methods could occupy the middle layer, and any number of visualization and analysis applications can interact with the data and middle layer elements. In fact, this is the goal: no one analysis application is likely to meet the workflow needs of all users, and we shouldn't expect it to. The distributed infrastructure makes it possible for many users with different analysis or visualization needs to utilize common lower-level resources.

---

## Author Response (AR1)

Editors,

Attached please find a revised version of the manuscript 'Technical note: A prototype transparent-middle-layer data management and analysis infrastructure...'

This page summarizes the revisions. Note that page and line numbers are indexed to the revised manuscript, not the latexdiff output that follows in this file.

Overall, reviews of this manuscript were generally supportive of publication, but had relatively few suggestions for specific revisions to the paper and instead focused on broad discussion of aspects of geochronology data management that are important and interesting, but not directly related to the paper. I responded to the broader discussion topics at length in the author's responses to the reviews, but here I focus on the specific suggested revisions to the paper.

Reviewer Sebastian Kreutzer made the following suggestions:

1. A number of typographical and grammar corrections. These have been corrected in the revised manuscript, except that the issue of whether the name of an isotope should appear first fully written out (e.g., beryllium-10) or abbreviated ($^{10}$Be) is a matter of journal formatting and I have left that to be decided by the copy editors.

2. A request to clarify the definition of "middle-layer." I added some clarification to section 1 (p. 1, line 16 - p. 2, line 6 in revised version) and section 3 (p. 4, line 22-24 in revised version).

3. A request for more discussion of how important changes to the middle-layer calculations are to the results of exposure-dating studies. I added some discussion of this in section 2 (near p. 3, line 5 in revised version)

4. A request to discuss the "FAIR data principles" in this paper. I responded to this request at length in the online discussion and argued that this was off topic with respect to the paper and that it brought up a number of marginally related issues that many researchers have strong opinions on, and that in my opinion would distract from the main point of the paper. Thus, I have not added this material. Note that in the open review model, even if this is not addressed in the text, a thorough airing of this issue is available to readers in the interactive online discussion.

Reviewer Richard Jones did not suggest any specific revisions to the paper. However, his comments indicated that there were areas of the paper that needed to be clarified. Specifically, I have tried to clarify and add additional discussion of the important point that the "ICE-D" prototype gives only one non-unique example of the elements of a transparent-middle-layer architecture, and many other software elements could fill the same roles (throughout sections 4.2 - 4.3 on pp. 6-8 of the revised version).

[revised manuscript text omitted]